# Feasibility and Acceptability of a Deep-Learning-Based Nipple Trauma Assessment System for Postpartum Breastfeeding Support

**DOI:** 10.3390/healthcare13172091

**Published:** 2025-08-22

**Authors:** Maya Nakamura, Hiroyuki Sugimori, Yasuhiko Ebina

**Affiliations:** 1Graduate School of Health Sciences, Hokkaido University, Sapporo 060-0812, Japan; 2Faculty of Health Sciences, Hokkaido University, Sapporo 060-0812, Japan; sugimori@hs.hokudai.ac.jp (H.S.); ebiyas@hs.hokudai.ac.jp (Y.E.)

**Keywords:** artificial intelligence, breastfeeding, deep learning, maternal satisfaction, nipple trauma, postpartum support

## Abstract

**Background/Objectives**: Nipple trauma is a common challenge during the early postpartum period, often undermining maternal confidence and breastfeeding success. Although deep-learning-based image analysis offers the potential for objective and remote assessments, its feasibility in clinical practice has not been well examined. This study aimed to evaluate the feasibility and acceptability of a deep-learning-based nipple trauma assessment system and explore maternal perceptions of the intervention. **Methods**: A quasi-experimental study was conducted at a maternity hospital in Japan. Participants were assigned to intervention or control groups based on their delivery month. Mothers in the intervention group used a dedicated offline smartphone to photograph their nipples during hospitalization. Images were analyzed using a pretrained deep-learning model, and individualized feedback was delivered via a secure messaging platform. Self-administered questionnaires were collected at three points: late pregnancy, during hospitalization, and one month postpartum. Maternal experiences and satisfaction with breastfeeding were also assessed. **Results**: A total of 23 participants (intervention = 8 and control = 15) completed the study. The system functioned without technical errors, and no adverse events were reported. Most participants found the AI results useful, with 75% receiving high-confidence outputs (predicted class probability ≥ 60%). Participants expressed interest in real-time feedback and post-discharge use. Breastfeeding self-efficacy scores (BSES-SF) improved more in the intervention group (+9.8) than in the control group (+7.8). **Conclusions**: This study confirmed the feasibility and acceptability of a deep-learning-based nipple trauma assessment system during postpartum hospitalization. The system operated safely and was well received by participants. Future developments should prioritize real-time, remote functionality to support diverse maternal needs.

## 1. Introduction

Breastfeeding in the early postpartum period is widely recognized as critical to both maternal and infant health; yet, it is also a time when mothers face significant challenges. Among these, nipple trauma is one of the most common issues, reported in over 60% of breastfeeding women [1]. Painful nipple injuries can undermine maternal confidence and contribute to the premature cessation of breastfeeding [2,3]. Despite the importance of early support, in-person care is often limited during the first postpartum month due to healthcare system constraints and social factors such as nuclear family structures [4].

In Japan, for instance, although nearly all mothers express an intention to breastfeed, the exclusive breastfeeding rate drops to just over 50% by one month postpartum [5]. A shortage of skilled lactation support, especially among midwives with increasing responsibilities, exacerbates this problem. Technological interventions may help bridge this gap. According to the Technology Acceptance Model (TAM), user acceptance of technology is influenced primarily by perceived usefulness and perceived ease of use, which are crucial factors in ensuring the successful adoption of AI-based health interventions [6]. Since the COVID-19 pandemic, the expansion of online support tools has shown potential to enhance maternal confidence and improve breastfeeding outcomes [7].

Recent advances in artificial intelligence (AI), particularly deep learning, have made it possible to automate the classification of medical images with high accuracy [8,9]. In a previous study, we developed a deep-learning-based system that analyzed photographs of the nipple taken by postpartum mothers and automatically classified the degree of nipple trauma using convolutional neural networks (CNNs) trained on clinical image data [10]. This system offered objective assessments, which can complement clinical judgments and reduce variability in caregiver evaluations [11].

While the system’s technical validity has been demonstrated in prior research, its practical feasibility and acceptability in real-world clinical settings remain unclear. As digital-native generations of mothers become more prevalent, it is essential to understand how such AI-assisted tools are received by users and whether they can be safely and effectively integrated into postpartum care.

Therefore, the aim of this study was to evaluate the feasibility and acceptability of a deep-learning-based nipple trauma assessment system implemented during postpartum hospitalization. In addition, maternal perceptions of the intervention, including subjective experiences and breastfeeding self-efficacy, were explored.

## 2. Materials and Methods

### 2.1. Study Design and Setting

The study procedures were structured into three sequential phases for clarity:

Phase 1: Preparation and Recruitment (Late Pregnancy)—Participants in the intervention group (expected delivery in August 2024) were recruited in late pregnancy (July 2024), and those in the control group (expected delivery in September 2024) were recruited in late pregnancy (July–August 2024). Eligibility was confirmed according to the inclusion and exclusion criteria, and baseline data (breastfeeding intention, demographics, and obstetric history) were collected via an online questionnaire.

Phase 2: Intervention and Inpatient Data Collection—During postpartum hospitalization (August–September 2024 for the intervention group and September–October 2024 for the control group), participants in the intervention group photographed their nipples using a dedicated offline smartphone. Images were analyzed using the deep-learning system, and individualized feedback was provided via the secure LINE account. Control group participants received standard postpartum care only.

Phase 3: Follow-Up (One Month Postpartum)—Follow-up assessments were conducted at the routine one-month postpartum check-up (September–October 2024 for the intervention group and October–November 2024 for the control group). Participants completed the follow-up questionnaire assessing breastfeeding self-efficacy, satisfaction, and (for the intervention group) perceptions of the AI system.

Thus, although the intervention period for each participant spanned from late pregnancy (T1) to one month postpartum (T3), all recruitment and data collection occurred between August and November 2024. To minimize bias from caregiver variability, only one facility was used.

### 2.2. Participants

Inclusion criteria were Japanese women aged ≥18 years in the late stage of pregnancy and intending to breastfeed.

Exclusion criteria were cases where direct breastfeeding of the newborn was not assured (e.g., maternal medication for mental illness, risk of mother–infant infection, and anticipated long-term separation from the infant) or where general smartphone operation was difficult.

### 2.3. Intervention

Participants in both groups received standard postpartum care, including lactation support during hospitalization and home visits. In addition, the intervention group received feedback based on a deep-learning-based assessment of nipple images.

During hospitalization, mothers in the intervention group used a dedicated, offline smartphone to take close-up photographs of their nipples. These images were then analyzed using the deep-learning system, which classified nipple conditions into “mild,” “moderate,” or “severe” categories. Feedback and breastfeeding advice were provided through the official research LINE account based on the system’s results. To assess user impressions, participants in the intervention group were asked to complete four Likert-scale items and one open-ended question.

The smartphone model used was Black view A95, with a resolution of 720 × 1600 pixels. All images were collected and analyzed in an offline environment to ensure data security.

### 2.4. Researcher Reflexivity

The first author (MN) was a certified midwife with over 15 years of clinical experience in postpartum care, including management of nipple trauma. This professional background informed the design of the intervention, choice of outcome measures, and interpretation of nipple trauma severity. To reduce potential bias, standardized procedures for image analysis and feedback delivery were applied, and data interpretation was guided by predefined coding criteria. Quantitative analyses were independently verified by the second and third authors, both of whom had expertise in maternal health research and AI-assisted medical imaging, to ensure consistency and minimize subjective influence.

### 2.5. Deep-Learning Model Architecture and Performance

The deep-learning-based nipple trauma assessment system used in this study was originally developed and validated following the protocol described in our previous work (Nakamura et al., 2025) [10]. The model consisted of two stages:Object detection—Using a YOLOv2 architecture with a ResNet50 backbone to identify the nipple and areola regions. Images were resized to 512 × 512 pixels, with bounding box coordinates adjusted accordingly and non-maximum suppression applied at an IoU threshold of 0.5. The object detector achieved high accuracy (mean average precision, AP@0.5: nipple = 0.93, areola = 0.92) and processed approximately 50.7 frames per second.Image classification—The cropped nipple regions were classified into “None,” “Minor,” “Moderate,” or “Severe” trauma categories using a ResNet50-based convolutional neural network fine-tuned from ImageNet-pretrained weights. As reported in our prior study, performance metrics from five-fold cross-validation included average AUC values of 0.78 for severe trauma and an overall classification accuracy of 0.50 for the four-class model.

The classification threshold for high-confidence output in the present study was retained at ≥0.60 predicted class probability, consistent with our prior work. The model was implemented in MATLAB R2023b (MathWorks, Natick, MA, USA) and trained using NVIDIA RTX A6000 GPUs (NVIDIA, Santa Clara, CA, USA).

### 2.6. Data Collection

Figure 1 shows the overview of the study protocol. Three self-administered questionnaires were distributed via Google Forms through the LINE platform at three time points:T1 (Late pregnancy): breastfeeding intention, maternal demographics, and obstetric history.T2 (Hospitalization postpartum): delivery details, feeding method, breastfeeding self-efficacy (BSES-SF) [11,12], and breastfeeding satisfaction.T3 (One-month postpartum): intervention feedback (intervention group only), current feeding method, breastfeeding self-efficacy (BSES-SF), and breastfeeding satisfaction.

### 2.7. Measures

Impressions of using the deep-learning system were assessed among participants in the intervention group using four single-item, 5-point Likert scales.

Usefulness evaluations: Participants rated the usefulness of the AI-generated image classification results and the helpfulness of the feedback content. Response options ranged from 1 (“Helpful”) to 5 (“Not helpful”).Recommendation intent: Participants were asked whether they would recommend the system to family or friends if it was free of charge and if real-time feedback were available. Response options ranged from 1 (“Would recommend”) to 5 (“Would not recommend”).Breastfeeding Self-Efficacy Scale—Short Form (BSES-SF): A 14-item, 5-point Likert scale adopted from the validated Japanese version [12] through a standardized forward–backward translation procedure involving experts in maternal health and psychometrics. A score of 50 or below was considered indicative of low breastfeeding confidence [13].Breastfeeding satisfaction: A single-item, 3-point Likert scale assessing overall satisfaction with the current breastfeeding experience. Participants selected one of the following responses: “Satisfied,” “Somewhat satisfied,” or “Not satisfied.”

### 2.8. Ethical Considerations

The study was approved by the Ethics Committee of the Faculty of Health Sciences, Hokkaido University (Approval No. 24-18). Prior to participation, all participants received detailed explanations about the study’s purpose, procedures, and data handling, including assurances that participation was voluntary and that refusal or withdrawal would not result in any disadvantage. Written informed consent was obtained from all participants who agreed to join the study.

All data were anonymized and managed securely. Nipple images were captured using a dedicated offline device, with no identifiable information stored alongside the images. The images were transferred directly to a secure offline computer without any transmission via the internet, ensuring strict protection of participant privacy and minimizing risk of data leakage throughout the study.

## 3. Results

### 3.1. Participant Flow

Figure 2 shows the study flow. A total of 24 participants consented to join the study, with 8 in the intervention group (expected delivery in August 2024) and 16 in the control group (expected delivery in September 2024). One control participant did not complete the final questionnaire at the one-month check-up and was excluded. Thus, 23 participants (intervention group: *n* = 8; control group: *n* = 15) were included in the final analysis. The complete response rate was 95.8%.

### 3.2. Participant Characteristics

Table 1 shows the demographic characteristics of the participants. The majority were primiparous (intervention group: 62.5%; control group: 80.0%). Approximately 75% of all participants were employed. There were no significant differences between the groups except for age; the control group was significantly younger (*p* < 0.01). All participants delivered at term, with a mean gestational age of approximately 39 weeks. Most births were vaginal, with a few cesarean sections observed in the control group. Regarding breastfeeding intentions during pregnancy, participants in the control group were more likely to respond “Willing to breastfeed if milk is available” (*n* = 13, 86.6%). In contrast, in the intervention group, three participants (37.5%) selected either “I intend to use formula” or “No specific plan,” indicating a relatively lower baseline intention to exclusively breastfeed.

### 3.3. Analysis of AI Feedback and User Evaluation

Figure 3 presents an example interface displaying the AI analysis results, and Table 2 summarizes image-related information and the corresponding deep-learning outputs. The class-wise posterior probabilities for each image were visualized on the software. The posterior class probabilities returned by MATLAB’s predict function were displayed as a bar chart, with class labels on the x-axis and probability values (ranging from 0 to 1) on the y-axis. In the intervention group, all submitted images were successfully analyzed by the AI system without technical errors, and no adverse events were reported by participants.

Based on the system’s classification, 12.5% of participants received a “green” result, indicating no visible nipple damage; 25.0% received a “yellow” result, indicating mild damage; 62.5% received a “red” result, indicating severe damage.

Notably, the majority of participants (75%) received high-probability diagnostic outputs from the system (i.e., predicted class probability ≥ 60%). This suggests that the AI system not only functioned reliably but also provided results with a high level of certainty, potentially enhancing user trust in the feedback.

Figure 4 presents the mothers’ feedback regarding the AI system. Ratings on usefulness and recommendation intention were moderate (means around 2–3 on a 5-point scale). Two mothers provided open-ended comments, noting that the AI helped them recognize nipple trauma and made it easier to apply ointment effectively. Overall, participants rated the system as helpful (mean score for image judgment = 2.62; for feedback content = 2.75; on a 5-point Likert scale, where 1 = helpful and 5 = not helpful). In terms of perceived value, most participants indicated they would recommend the system to others if it were free (mean = 2.37) and even more favorably if real-time feedback were available (mean = 2.12).

In addition, open-ended responses were collected from two participants (25%) in the intervention group. One mother expressed a desire to receive support not only during hospitalization but also in the early postpartum period at home (e.g., 4 days to 2 weeks after birth). Another mother noted that identifying the exact location of nipple trauma made it easier to apply ointment appropriately, especially when pain was widespread and the lower part of the nipple was difficult to visualize. These qualitative responses support the potential utility of AI-based image feedback for improving mothers’ self-care and access to timely support.

These findings highlight the system’s operational feasibility and suggest strong user interest in enhanced features such as real-time analysis and post-discharge usability.

### 3.4. Breastfeeding Status at Hospitalization and One-Month Check-Up

Table 3 summarizes changes in breastfeeding practices. At hospitalization, none of the mothers in the intervention group exclusively breastfed, whereas 13.3% in the control group did. At one month postpartum, the percentage of mothers exclusively breastfeeding increased in both groups, with 46.6% in the control group and 12.5% in the intervention group.

### 3.5. Maternal Perception of Breastfeeding

Table 4 presents maternal evaluation scores. At hospitalization, 66.6% of the control group reported being satisfied with their feeding method, compared to only 25.0% in the intervention group. By one month, 62.5% of the intervention group also reported satisfaction.

The BSES-SF scores increased in both groups over time. Although the differences were not statistically significant, the control group consistently showed higher mean scores. The mean score increased by about 10 points in the intervention group and 8 points in the control group.

## 4. Discussion

This study explored the feasibility and acceptability of a deep-learning-based nipple trauma assessment system implemented during postpartum hospitalization. Despite the small sample size, the findings offer important insights into the system’s potential clinical application and user perception.

### 4.1. Participant Characteristics and Baseline Trends

The characteristics of participants in this study were generally consistent with previous reports on Japanese postpartum women [14]. Most participants were primiparous, employed, and part of nuclear families, reflecting contemporary maternal demographics in Japan. Although the control group was significantly younger, all participants were of the digital-native generation, and the influence of age differences was considered minimal.

However, previous research has suggested that older maternal age may be associated with higher breastfeeding satisfaction [15]. Therefore, age should be taken into consideration when interpreting the results.

### 4.2. System Feasibility and Safety

The AI system was successfully implemented without technical errors or adverse events, supporting its feasibility for clinical use during postpartum hospitalization. All images submitted by mothers were correctly analyzed, and diagnostic results were consistent with previously reported distributions of nipple trauma severity in postpartum women [16]. Notably, 75% of the intervention group received high-probability outputs (≥60%), suggesting the system’s robustness and potential to inspire user confidence. This finding is consistent with the Technology Acceptance Model (TAM), which emphasizes that factors such as perceived ease of use and confidence—reflected here as diagnostic certainty—play key roles in user acceptance of health technologies [6]. Overall, these results support the effective integration of AI-based assessments into real-world maternity care within secure, offline environments.

### 4.3. Maternal Feedback and Usability

Overall, participants rated the system as helpful, and many expressed interest in using similar tools post-discharge. Ratings on usefulness and recommendation intent were moderate to favorable, with lower scores (i.e., stronger interest) reported for a version offering real-time feedback. These findings are consistent with earlier reports indicating that timely, interactive support enhances breastfeeding satisfaction and self-efficacy [17].

Open-ended responses further supported the system’s value: one mother appreciated the ability to locate trauma sites for accurate ointment application, while another requested continued support during the early post-discharge period. These responses suggest that AI-based visual feedback may empower mothers to manage nipple care more independently, especially when professional support is not readily accessible.

### 4.4. Impact on Breastfeeding Experience

While no statistically significant group differences were observed, improvements in breastfeeding self-efficacy (BSES-SF) were noted in both groups, with a slightly greater increase in the intervention group. Although the intervention was limited to inpatient care, these trends support the hypothesis that even minimal, individualized AI-assisted feedback may contribute to maternal confidence and satisfaction, echoing findings from earlier research on breastfeeding support interventions [7]. Notably, the intervention group started with lower prenatal breastfeeding intention and satisfaction during hospitalization, potentially influencing post-intervention outcomes. These baseline differences should be considered when interpreting the results.

Importantly, due to the lack of a security system for transmitting sensitive images from home, the intervention was conducted during hospitalization rather than after discharge, as originally planned. Ideally, the impact of such an AI-based system should be evaluated during the post-discharge period, when professional support is limited and mothers are managing breastfeeding more independently. Therefore, the observed changes in self-efficacy and satisfaction should be regarded as secondary outcomes, reflecting the feasibility and preliminary effects of inpatient implementation, rather than the full potential of post-discharge intervention.

### 4.5. Practical and Ethical Challenges

Several practical and ethical challenges emerged during the implementation of this study. Initially, we aimed to enable prompt AI-based diagnosis by allowing participants to submit images remotely, including after hospital discharge. However, we were compelled to revise our study protocol because no image transmission system currently available in Japan met the necessary standards for privacy, security, and user confidence. As a result, several limitations arose. First, the time lag (0–2 days) between image submission and feedback delivery limited the immediacy of support. Second, the use of sensitive images may have discouraged some potential participants, particularly multiparous women. In future studies, ensuring secure yet user-friendly remote data transmission will be essential, especially for post-discharge use. Real-time feedback functionality, end-to-end encryption, and automated anonymization could enhance both effectiveness and acceptability.

Moreover, the number of participants was substantially lower than originally planned, particularly among multiparous women. This limited recruitment may have reduced the diversity of maternal needs captured by the study. To enhance the appeal and practicality of the system for a broader range of mothers, future iterations should consider expanding its functionality beyond nipple trauma—for example, to include support for managing mastitis, guidance on formula supplementation, and general breastfeeding consultation. Such enhancements may increase the system’s relevance, particularly for experienced mothers who may have different support expectations and preferences.

### 4.6. Clinical Implications and Future Directions

The present study demonstrates that a deep-learning-based nipple trauma assessment system can be safely and feasibly implemented during postpartum hospitalization. While the intervention was limited to a single image submission during inpatient care, the results suggest that even minimal AI-assisted feedback can positively influence maternal perceptions of breastfeeding. This is particularly relevant in modern maternity care contexts, where staffing shortages and shorter hospital stays often reduce opportunities for in-person breastfeeding support.

The integration of AI into postpartum care offers a promising strategy to supplement midwife-led interventions, especially during the critical transition from hospital to home. Given that many participants expressed interest in continued support after discharge, future iterations of the system should prioritize real-time functionality and remote usability. These advancements will require the development of secure image transfer protocols, user-friendly mobile applications, and ethical frameworks for handling sensitive health data. Importantly, such features must be designed to align with maternal preferences for privacy, usability, and timeliness.

In addition, this study highlights the need to diversify and strengthen recruitment strategies. The underrepresentation of multiparous women suggests that broader system appeal—through expanded features such as mastitis detection or personalized feeding advice—may be necessary to engage a wider demographic. Tailoring AI feedback to maternal experience levels, breastfeeding goals, and psychosocial context could improve both uptake and sustained use.

Finally, future research should aim for larger, more diverse samples and longitudinal designs to assess the long-term impact of AI-assisted support on breastfeeding duration, maternal confidence, and infant health outcomes. As the population of digital-native mothers continues to grow, the integration of AI into maternal health services must evolve in parallel, ensuring accessibility, equity, and clinical relevance across care settings.

### 4.7. Limitations

The small sample size (n = 23) in the present feasibility study limits the generalizability of findings and precludes a robust statistical comparison between groups. While the underlying deep-learning model was trained and validated on an augmented dataset of 753 clinical images in previous research, the current clinical application was tested with a limited number of participants and a single image submission per mother. Future studies should incorporate larger, more diverse postpartum populations and evaluate the model with external datasets to strengthen reproducibility and scientific validity.

## 5. Conclusions

This study demonstrated the feasibility and acceptability of a deep-learning-based nipple trauma assessment system implemented during postpartum hospitalization. The system functioned safely and reliably for all participating mothers, with no adverse events reported. Most participants found the feedback helpful, and many expressed interest in future use beyond hospitalization.

To enhance clinical applicability, future developments should focus on secure remote access, real-time analysis, and expanded features to support broader breastfeeding concerns. These improvements may increase usability and meet the diverse needs of postpartum mothers.

In summary, AI-assisted image analysis shows promise as a supportive tool in maternal care and warrants further research for broader implementation.

## Figures and Tables

**Figure 1 healthcare-13-02091-f001:**
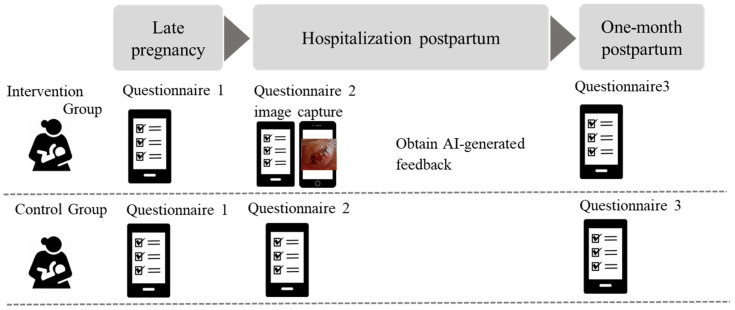
Overview of the study protocol.

**Figure 2 healthcare-13-02091-f002:**
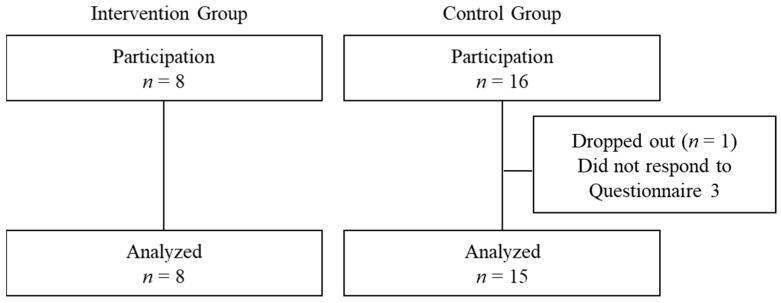
Study flow.

**Figure 3 healthcare-13-02091-f003:**
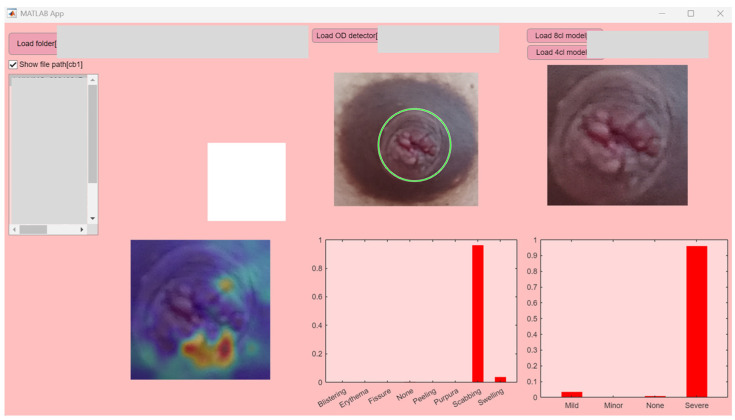
Example interface showing the analysis result.

**Figure 4 healthcare-13-02091-f004:**
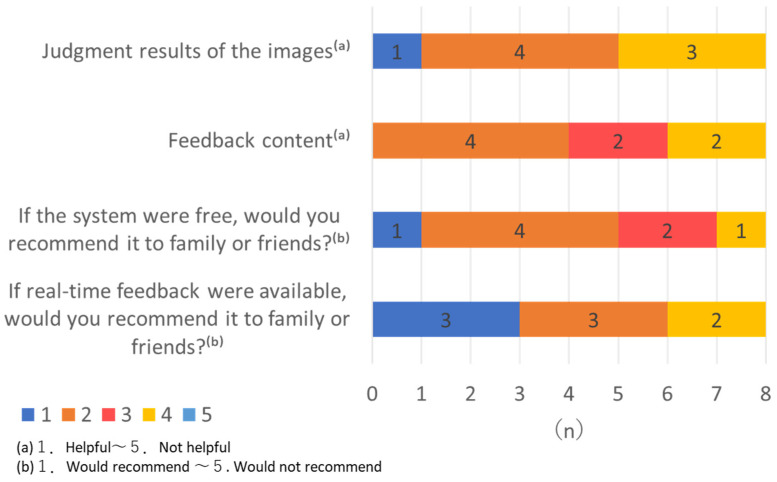
Mothers’ feedback regarding the AI system.

**Table 1 healthcare-13-02091-t001:** Demographic and clinical characteristics of mothers.

Characteristic	Intervention Group (n = 8) n (%)	Control Group (n = 15) n (%)	χ^2^/t	*p*-Value
Primiparous	5 (62.5%)	12 (80.0%)	0.29	
Age (years), mean ± SD	34.1 ± 3.09	29.2 ± 2.93		<0.01
Currently employed	6 (75.0%)	11 (73.3%)	1	
Financial concerns	2 (25.0%)	5 (33.3%)	1	
Living with others besides husband/children	0 (0.0%)	1 (6.6%)	1	
Gestational diabetes or threatened preterm labor	1 (12.5%)	2 (13.3%)	1	
Breastfeeding experience			0.46	
No prior experience	5 (62.5%)	12 (80.0%)		
Exclusive breastfeeding	1 (12.5%)	2 (13.3%)		
Mixed feeding	2 (25.0%)	1 (6.6%)		
Formula only	0 (0.0%)	0 (0.0%)		
Intention to breastfeed during pregnancy			0.09	
Strong desire to breastfeed	1 (12.5%)	2 (13.3%)		
Willing to breastfeed if milk is available	4 (50.0%)	13 (86.6%)		
I intend to use formula	1 (12.5%)	0 (0.0%)		
No plan	2 (25.0%)	0 (0.0%)		
Gestational age at birth (weeks), mean ± SD	39.1 ± 1.24	39.0 ± 1.03		0.98
Delivery mode			0.49	
Vaginal delivery without anesthesia	4 (50.0%)	8 (53.3%)		
Painless delivery (epidural, etc.)	4 (50.0%)	5 (33.3%)		
Cesarean section	0 (0.0%)	2 (13.3%)		

**Table 2 healthcare-13-02091-t002:** Image-related information and deep-learning analysis results.

Participant No.	Analysis Result	Probability of Result	Nipple	Postpartum Day of Imaging	Days Required for Feedback
1	Red	90–100%	Left	Day 4	0 days
2	Red	100%	Left	Day 3	0 days
3	Green	40–50%	Right	Day 2	2 days
4	Yellow	40–50%	Right	Day 1	1 day
5	Red	90–100%	Left	Day 4	2 days
6	Red	70–80%	Left	Day 2	2 days
7	Red	80–90%	Left	Day 1	1 day
8	Yellow	60–70%	Right	Day 2	2 days

**Table 3 healthcare-13-02091-t003:** Breastfeeding progress during hospitalization and at the one-month postpartum check-up.

	During Hospitalization				One-Month Check-Up			
	Intervention Group (n = 8) n (%)	Control Group (n = 15) n (%)	χ^2^	*p*	Intervention Group (n = 8) n (%)	Control Group (n = 15) n (%)	χ^2^	*p*
Feeding method within 24 h			4.14	0.13			3.65	0.09
- Exclusively breastfeeding	0 (0.0%)	2 (13.3%)			1 (12.5%)	7 (46.6%)		
- Mixed feeding (mostly breast milk)	5 (62.5%)	12 (80.0%)			6 (75.0%)	5 (33.3%)		
- Mixed feeding (mostly formula)	3 (37.5%)	1 (6.6%)			1 (12.5%)	2 (13.3%)		
- Formula only	0 (0.0%)	0 (0.0%)			0 (0.0%)	1 (6.6%)		

**Table 4 healthcare-13-02091-t004:** Mothers’ self-evaluation of breastfeeding during hospitalization and one-month check-up.

	During Hospitalization				One-Month Check-Up			
	Intervention Group (n = 8) n (%)	Control Group (n = 15) n (%)	χ^2^	*p*	Intervention Group (n = 8) n (%)	Control Group (n = 15) n (%)	χ^2^	*p*
Perception of the breastfeeding method			0.03				1	
Satisfied	2 (25.0%)	10 (66.6%)			5 (62.5%)	11 (73.3%)		
Somewhat satisfied	3 (37.5%)	5 (33.3%)			2 (25.0%)	4 (26.6%)		
Not satisfied	3 (37.5%)	0 (0.0%)			0 (0.0%)	0 (0.0%)		
BSES-SF	31.0 ± 4.34	38.8 ± 12.47		0.06	40.87 ± 11.24	46.66 ± 12.94		0.36

## Data Availability

The raw data supporting the conclusions of this article will be made available by the authors on request.

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
