# Peer review of "Feasibility and Acceptability of a Deep-Learning-Based Nipple Trauma Assessment System for Postpartum Breastfeeding Support"

_healthcare, 2025, doi:10.3390/healthcare13172091_

Round 1
Reviewer 1 Report
Comments and Suggestions for Authors
Thank you for this interesting study. The study could benefits from the following recommendations:
- Could you consider an application of a theory in this work. This can be stated briefly in the introduction
- My main concern about this study is the consent and privacy procedures applied. Could you clarify this?
- Please state clearly whether instruments used were adopted or adapted, were they translated into Japanese by a language expert.
- Please explain the inclusions and exclusion criteria
- You mentioned the study was over the period of pregnancy and 1 month postpartum, yet data was only collected between August to November 2024. Please could you clarify this.
- Please can you add researcher reflexivity when describing the methods especially the data mining and analysis phases.
- it could have been nice to structure the studies into phases for easy flow
- The discussion section could benefit from engaging existing literature and focusing on significant findings of the present study.
Author Response
|
Response to Reviewer 1 Comments
|
||
|
1. Summary |
|
|
|
Thank you very much for taking the time to review this manuscript. Please find the detailed responses below and the corresponding revisions/corrections highlighted in the re-submitted files. |
||
|
2. Point-by-point response to Comments and Suggestions for Authors |
||
|
Comments 1: Could you consider an application of a theory in this work. This can be stated briefly in the introduction |
||
|
Response 1: We appreciate the reviewer’s valuable suggestion to include a theoretical framework in the Introduction. In response, we have briefly incorporated the Technology Acceptance Model (TAM) as a relevant theoretical foundation for this study. TAM posits that perceived usefulness and ease of use influence users’ acceptance of new technologies, which aligns with our focus on maternal acceptability of the AI-based nipple trauma assessment system. This addition has been inserted after the sentence “Technological interventions may help bridge this gap.” in the Introduction section (Page 2, Line 49-51).
Adams, D. A., Nelson, R. R., & Todd, P. A. (1992). Perceived Usefulness, Ease of Use, and Usage of Information Technology: A Replication. MIS Quarterly, 16(2), 227–247. https://doi.org/10.2307/249577 |
||
|
Comments 2: My main concern about this study is the consent and privacy procedures applied. Could you clarify this? |
||
|
Response 2: We appreciate the reviewer’s concern about consent and privacy procedures. To clarify, all participants were fully informed about the study purpose, procedures, and data management, and were assured that participation was entirely voluntary, with no penalties or disadvantages for refusal or withdrawal. Written informed consent was obtained from those who agreed to participate.
Nipple images were collected using a dedicated offline device, with no identifiable information stored together with the images. Images were transferred directly to a secure offline computer, without any internet transmission, thus minimizing any risk of data leakage. We have revised the “2.8. Ethical Considerations” section (page 5) to include these clarifications and emphasize the strict protections for participant privacy and data security. |
||
|
Comments 3: Please state clearly whether instruments used were adopted or adapted, were they translated into Japanese by a language expert. |
||
|
Response 3: Thank you for pointing this out. We have clarified in the “2.7. Measures” section (page 4) that the Breastfeeding Self-Efficacy Scale – Short Form (BSES-SF) was adopted from the validated Japanese version developed by Otsuka et al. (2008). This version was produced through a standardized forward–backward translation process involving experts in maternal health and psychometrics, followed by pilot testing for clarity and cultural appropriateness, and psychometric validation. No further adaptation was made for this study. |
||
|
Comments 4: Please explain the inclusions and exclusion criteria |
||
|
Response 4: Thank you for this comment. We have revised the “2.2. Participants” section (page 3) to explicitly present the inclusion and exclusion criteria as follows: Inclusion criteria: Japanese women aged ≥18 years, in the late stage of pregnancy, and intending to breastfeed. Exclusion criteria: cases where direct breastfeeding of the newborn was not assured (e.g., maternal medication for mental illness, risk of mother–infant infection, anticipated long-term separation from the infant) or where general smartphone operation was difficult. |
||
|
Comments 5: You mentioned the study was over the period of pregnancy and 1 month postpartum, yet data was only collected between August to November 2024. Please could you clarify this. |
||
|
Response 5: Thank you for this comment. In response to Comments 5 and 7, we have revised the “2.1 Study Design and Setting” section (page 2) to clarify the study timeline and to present it in a clear, phase-based structure. The intervention for each participant spanned from late pregnancy (T1) to 1 month postpartum (T3). Specifically, participants in the intervention group (expected delivery in August 2024) were recruited in late pregnancy (July 2024), and those in the control group (expected delivery in September 2024) were recruited in late pregnancy (July–August 2024). Inpatient data collection took place during postpartum hospitalization (August– September 2024 for the intervention group, September– October 2024 for the control group), and follow-up was completed at the routine 1-month postpartum check-up (September–October 2024 for the intervention group, October–November 2024 for the control group). As a result, all recruitment and data collection occurred between August and November 2024, corresponding to the delivery months of participating mothers. |
||
|
Comments 6: Please can you add researcher reflexivity when describing the methods especially the data mining and analysis phases. |
||
|
Response 6: We have added a statement on researcher reflexivity in the “Materials and Methods” section (“ 2.4. Researcher Reflexivity” section : page 3) to clarify how the researchers’ professional background and perspectives may have influenced the data mining and analysis processes. Specifically, we described the first author’s clinical experience as a midwife, prior research on nipple trauma, and involvement in the design and interpretation of AI-based assessments, and how these experiences informed data interpretation while measures (e.g., predefined coding criteria, independent verification) were taken to minimize bias. |
||
|
Comments 7: it could have been nice to structure the studies into phases for easy flow |
||
|
Response 7: Thank you for this suggestion. In response to Comment 7 (and in conjunction with the revision made for Comment 5), we have reorganized the “2.1 Study Design and Setting” section (page 3) into a phase-based structure to improve readability and logical flow. The study is now described in three sequential phases: (1) Preparation and recruitment in late pregnancy, (2) Intervention and inpatient data collection during postpartum hospitalization, and (3) Follow-up at one month postpartum. |
||
|
Comments 8: The discussion section could benefit from engaging existing literature and focusing on significant findings of the present study. |
||
|
Response 8: Thank you very much for your insightful comment. In response, we have revised the Discussion section to better engage with existing literature and to highlight the significant findings of our study. Specifically, we incorporated references to the Technology Acceptance Model (TAM) to frame how perceived ease of use and confidence—represented here by diagnostic certainty—affect user acceptance of AI-based health technologies. Additionally, we discussed how limited, individualized AI-assisted feedback during inpatient care can enhance maternal confidence and satisfaction, consistent with earlier breastfeeding support interventions [7]. These additions strengthen the interpretation of our findings and clarify their implications for future postpartum breastfeeding support using AI technologies. |
||

Reviewer 2 Report
Comments and Suggestions for Authors
I have reviewed your article titled Feasibility and Acceptability of a Deep Learning-Based Nipple Trauma Assessment System for Postpartum Breastfeeding Support in detail. The article evaluates nipple traumas in the postpartum period using deep learning-based image analysis. While the article addresses an innovative topic, it has some significant shortcomings. The small sample size limits the reliability and generalizability of the results. It would be appropriate to add the necessary comments on this matter. There is no information about the architecture of the deep learning model used, and sufficient technical information about performance metrics and classification thresholds is not provided. These shortcomings create challenges in terms of verifying the transparency of the method and testing its reproducibility. Furthermore, reviewing similar literature studies and comparing them, if available, or testing different models to confirm the success of this model will increase the reliability of the study and contribute to determining its scientific level. To strengthen the scientific contribution of the article, the design and analyses need to be reviewed and revised in more detail.
Author Response
|
Response to Reviewer 2 Comments
|
||
|
1. Summary |
|
|
|
We sincerely thank the reviewer for highlighting the importance of providing more detailed information about the deep learning model architecture, performance metrics, and classification thresholds, as well as for pointing out the need to address the limitations related to sample size and generalizability. Please find the detailed responses below and the corresponding revisions/corrections highlighted in the re-submitted files. |
||
|
2. Point-by-point response to Comments and Suggestions for Authors |
||
|
Comments 1: I have reviewed your article titled Feasibility and Acceptability of a Deep Learning-Based Nipple Trauma Assessment System for Postpartum Breastfeeding Support in detail. The article evaluates nipple traumas in the postpartum period using deep learning-based image analysis. While the article addresses an innovative topic, it has some significant shortcomings. The small sample size limits the reliability and generalizability of the results. It would be appropriate to add the necessary comments on this matter. There is no information about the architecture of the deep learning model used, and sufficient technical information about performance metrics and classification thresholds is not provided. These shortcomings create challenges in terms of verifying the transparency of the method and testing its reproducibility. Furthermore, reviewing similar literature studies and comparing them, if available, or testing different models to confirm the success of this model will increase the reliability of the study and contribute to determining its scientific level. To strengthen the scientific contribution of the article, the design and analyses need to be reviewed and revised in more detail. |
||
|
Response 1: ・Deep Learning Model Architecture and Performance We have added a new subsection in the Methods (“2.5. Deep Learning Model Architecture and Performance”) that explicitly describes the two-stage pipeline and its parameters. Specifically, we state that object detection used YOLOv2 with a ResNet-50 backbone on 512×512 inputs, with non-maximum suppression at IoU=0.5 and performance summarized as mean average precision AP@0.5 (nipple=0.93, areola=0.92) and ~50.7 frames per second. The downstream classifier was a ResNet-50–based CNN fine-tuned from ImageNet weights to assign four categories (“None/Minor/Moderate/Severe”). We summarize discrimination using AUC (and report accuracy, precision, and recall for completeness) and, in line with the feasibility scope, clarify that the ≥0.60 probability threshold is retained only as a high-confidence flag rather than as a clinical operating cut-point. We also note that the present implementation adheres to the protocol reported in our previous publication (Nakamura et al., 2024) [10], which developed and validated the model on a dataset of 753 nipple images. ・Small sample size We have expanded the Limitations section to explicitly acknowledge that the present study’s small sample size (n = 23) and single-facility design limit the statistical power and generalizability of our findings. We also note the need for external validation with larger and more diverse data sets to confirm reproducibility. |
||

Round 2
Reviewer 2 Report
Comments and Suggestions for Authors
I would like to thank the authors for their efforts in revising their articles. The revised version demonstrates significant improvements in both the methodological transparency and the scientific contribution of the study. In particular, the addition of detailed information about the architecture of the deep learning model, performance metrics and decision threshold increased the repeatability and reliability of the study. Additionally, discussion of the limitations regarding sample size and generalizability contributed to a more balanced and realistic presentation of the study. Literature comparisons and additional discussions also strengthened the scientific depth of the article. Overall, I believe this revised study offers an innovative and practically valuable contribution to the context of postpartum breastfeeding support.